# Evolutionary View on Lactate-Dependent Mechanisms of Maintaining Cancer Cell Stemness and Reprimitivization

**DOI:** 10.3390/cancers14194552

**Published:** 2022-09-20

**Authors:** Petr V. Shegay, Anastasia A. Zabolotneva, Olga P. Shatova, Aleksandr V. Shestopalov, Andrei D. Kaprin

**Affiliations:** 1Federal State Budget Institution National Medical Research Radiology Center of the Ministry of Healthcare of the Russian Federation, 2nd Botkinsky pas., 3, 125284 Moscow, Russia; 2Department of Biochemistry and Molecular Biology, Faculty of Medicine, Pirogov Russian National Research Medical University, st. Ostrovityanova, 1, 117997 Moscow, Russia; 3Faculty of Medicine, RUDN University, st. Miklukho-Maklaya, 6, 117198 Moscow, Russia; 4Dmitry Rogachev National Medical Research Center of Pediatric Hematology, Oncology and Immunology, Ministry of Health of the Russian Federation, st. Samora Mashela, 1, 117997 Moscow, Russia

**Keywords:** lactate, lactic acid, glycolysis, carcinogenesis, malignant tumors, evolutionary oncology

## Abstract

**Simple Summary:**

For a long-time lactic acid has been considered as a toxic end-product of metabolism that cause different adverse effects. However, in recent decades a plenty of studies has refuted this suggestion and revealed many functions of lactate in living organisms. Lactic acid may be considered as one of the most ancient metabolites with signaling function and high regulatory activity. Lactate regulates key metabolic processes such as proteins expression, cells’ differentiation, inflammatory response, as well as provides stemness and unlimited cell growth, which is used by malignant tumors for their growth. Although lactatemia, observed in many cancer diseases, is possibly associated with the activation of ancient evolutionary defense mechanisms, aimed at combating metabolic disorders, and which tumors began to use for their own purposes – the acquisition of stem properties, rapid proliferation, and metastasis. In our review we aimed to summarize the accumulated knowledge about the functions of lactate in the process of carcinogenesis and to consider the possible evolutionary significance of the Warburg effect.

**Abstract:**

The role of lactic acid (lactate) in cell metabolism has been significantly revised in recent decades. Initially, lactic acid was attributed to the role of a toxic end-product of metabolism, with its accumulation in the cell and extracellular space leading to acidosis, muscle pain, and other adverse effects. However, it has now become obvious that lactate is not only a universal fuel molecule and the main substrate for gluconeogenesis but also one of the most ancient metabolites, with a signaling function that has a wide range of regulatory activity. The Warburg effect, described 100 years ago (the intensification of glycolysis associated with high lactate production), which is characteristic of many malignant tumors, confirms the key role of lactate not only in physiological conditions but also in pathologies. The study of lactate’s role in the malignant transformation becomes more relevant in the light of the “atavistic theory of carcinogenesis,” which suggests that tumor cells return to a more primitive hereditary phenotype during microevolution. In this review, we attempt to summarize the accumulated knowledge about the functions of lactate in cell metabolism and its role in the process of carcinogenesis and to consider the possible evolutionary significance of the Warburg effect.

## 1. Introduction

In recent years, interest in malignant neoplasms as a biological phenomenon found in almost all multicellular species has increased, indicating its deep evolutionary roots dating back to the dawn of multicellularity. In 1929, Boveri suggested that malignant neoplasms represented a type of atavism or a reversion to a more primitive hereditary phenotype. In a series of research papers, Boveri’s idea of malignant tumors as a type of atavism was developed into a detailed theory of the onset and progression of neoplasms, the provisions of which are confirmed by phylostratigraphy, a statistical approach used to study the evolutionary age of genes involved in carcinogenesis [1]. Phylostratigraphic analysis originates from the pioneering work of Domazet-Loso [2]. The researchers determined the age of cancer-associated genes using a series of compilations of such sequences and found two peaks in which the representation of the genes was the highest compared to the age distribution of all other human genes. The first peak was observed in the era of unicellularity before the emergence of eukaryotes. The second peak was correlated with the emergence of multicellular organisms. The most important observation was that the genes associated with carcinogenesis and younger than ~400 million years were present to a much lower extent, which confirms the main thesis about the ancient evolutionary roots of malignantly transformed cells. In fact, genes responsible for cellular cooperation in multicellular organisms (e.g., signaling, adhesion, angiogenesis, and migration) are the first to be damaged during carcinogenesis, which leads to a loss of regulatory functions. Furthermore, the existence of a close relationship between carcinogenesis and the early stages of embryogenesis is generally recognized. Indeed, as the tumor progresses, its cells dedifferentiate in the direction of “stemness” and, in general, bear more resemblance to cells at an early stage of embryogenesis or unicellular forms [3].

A team of Australian researchers applied the phylostratigraphy method to RNA transcript sequencing data from The Cancer Genome Atlas for seven solid tumors divided into 16 age categories [4]. They found that genes that emerged during evolution in unicellular forms were overexpressed in human malignant tumors, while the expression of genes that appeared in the stage of the emergence of multicellularity, on the contrary, was suppressed. At the same time, the overexpression of genes associated with unicellularity was caused by a serious dysregulation of control structures that occurred during the evolutionary transition to multicellularity. It should be noted that this “atavism” cannot be regarded simply as a reprimitivization to unicellularity. Tumor cells are most likely to follow the path of reprogramming the links between the gene networks that control processes in unicellular organisms and those that control processes in multicellular organisms [4].

Anaerobic glycolysis arose in evolution, even before the appearance of eukaryotes and is thus one of the most ancient methods to supply energy to a cell [5]. Tumor cells effectively use glycolysis to obtain a large amount of lactate, a metabolite that serves not only as cellular fuel but also allows the reprogramming of cells in the tumor microenvironment in such a way as to create the most favorable conditions for the growth and metastasis of malignant cells [6]. The study of the role of lactate in the malignant transformation becomes even more relevant in the light of the “atavistic theory of carcinogenesis”, which suggests that tumor cells return to a more primitive hereditary phenotype during microevolution. In this review, we try to summarize the knowledge about the functions of lactate in cell metabolism and its role in the process of carcinogenesis and to consider the possible evolutionary significance of the Warburg effect.

## 2. Results

### 2.1. The Evolutionary Significance of the Intensification of Glycolysis

Another argument in favor of the “atavistic theory of carcinogenesis”, which we talked about above, is the fact that tumor cells use glycolysis, but not oxidative phosphorylation, as the main metabolic pathway for glucose utilization. One hundred years ago, in the 1920s, Otto H. Warburg, examining sections of tumor tissues, found that malignant cells predominantly convert glucose into lactate, even under conditions of adequate cellular oxygenation (i.e., in the absence of hypoxia) [7]. Warburg associated such a metabolic feature of tumor cells, which he called “aerobic glycolysis”, with damage to mitochondria and pointed out that such a change in metabolism is a key sign of malignant cell transformation [8]. However, the Warburg effect is currently associated with the metabolic reprogramming that accompanies tumor transformation and is necessary for the rapid growth and proliferation of tumor cells, maintaining their stem properties, avoiding an immune system response, and survival [9]. Thanks to numerous studies in recent years, the signaling and regulatory properties of lactate have become known, which to some extent reveal the mechanisms of cell malignancy and allow us to look at their evolutionary significance. The view of the role of lactate in cell metabolism has undergone significant changes in recent decades; from the misconception of lactate as a “metabolic waste” and a toxic end-product of glycolysis that must be neutralized, researchers have come to understand the critical role of lactate in metabolism and its regulation. Using positron emission tomography (PET) imaging or the ^13^C-glucose tracer [10], it was demonstrated that lactate moving between producers (drivers) and consumers (recipients) performs at least three key functions: 1) it is the main source of energy for mitochondrial respiration; 2) it is the main precursor for gluconeogenesis; and 3) it is a signaling molecule involved, among other things, in metabolic reprogramming [11]. At the same time, the signaling function of lactate is of greatest interest in terms of studying the mechanisms of malignant transformation. On the one hand, lactate, a metabolite of glycolysis, is one of the most ancient signaling molecules and performs regulatory functions in both eu- and prokaryotes [12]; on the other hand, the action of lactate is amazingly versatile. As described below, lactate is a key oncometabolite that provides energy for tumor cells, stimulates the acquisition of stem-like properties by tumor cells, induces the transformation of the tumor microenvironment, and thus is essential for the growth, invasion, and metastasis of transformed cells.

### 2.2. Lactic Acid Causes Acidification of the Extracellular Space of the Tumor and Suppresses Antitumor Immunity

A well-known hallmark of tumor cells is their ability to evade or block the immune response. In view of the relatively young evolutionary age of adaptive immunity (less than 500 million years), the theory of “carcinogenesis as a process of regression to ancestral forms” predicts that the adaptive immune response should be inhibited shortly after the initiation of carcinogenesis [13]. One of the mechanisms by which a tumor escapes an immune response is the creation of an acidic microenvironment, which contributes to the suppression of the inflammatory response and the activation of immune cells [14]. Lactate formed in large amounts in tumor cells (Warburg effect) is cotransported with protons (H^+^) from tumor cells through monocarboxylate transporters (MCT1 and MCT4), which, on the one hand, leads to lactate accumulation (up to 45 mM) and, on the other, reduces the pH of the tumor microenvironment [7]. In turn, acidification leads to the restriction of the production of interferon gamma (IFN-γ) by T cells that invade the tumor and prevents the activation of NK cells, which ultimately contributes to avoiding the immune response by the tumor and inhibits its growth [15]. Furthermore, various studies have shown that lactic acid can enhance the proinflammatory function of immune cells, the signaling pathway of toll receptor 4 (TLR 4), and nuclear factor (NF)-κB-dependent gene regulation [16]. High concentrations of lactate have been shown to stimulate anti-inflammatory M2 macrophage polarization through HIF1α stabilization or through epigenetic mechanisms such as histone lactylation [17]. Lactate also influences the phenotype and functionality of dendritic cells (DCs). Mechanistically, lactate may reduce the basal expression of CD1; promote maintaining a tolerogenic phenotype, characterized by reduced secretion of IL-12 and increased secretion of IL-10 in response to TLR stimulation; and the impair migratory response to chemokines [18]. Moreover, acidosis in the tumor microenvironment led to a decrease in the production of cytokines and to the loss of cytotoxic effector functions of T cells [19]. 

A high concentration of lactic acid in the tumor microenvironment disrupts the [H^+^] gradient between T cells and their environment, reducing the monocarboxylate transporter 1 (MCT1)-mediated export of lactic acid from T cells. This inhibits the proliferation of effector T cells. At the same time, the neutralization of the acidic environment, for example, using bicarbonate therapy, increases tumor infiltration by T cells, slows down its growth, and contributes to an increase in the effectiveness of therapy with immune checkpoint inhibitors (anti-PD1/PDL1 therapy) [14,20,21]. 

On the other hand, high concentrations of lactate in cells stimulate adenosine transporters, and adenosine is secreted into the extracellular space. In turn, the lactate anion activates the key enzyme of adenosine catabolism and thus prevents adenosinergic immunosuppression in the tumor microenvironment [21]. Therefore, the question remains whether lactate performs a protective function or is a factor of aggression in tumor growth.

### 2.3. Lactate Serves as a Fuel Molecule for Proliferating Cells

For many physiological and pathological processes, the functioning of the lactate shuttle mechanism is well-known, where lactate anions are exported from one cell and imported into another cell to serve as a substrate for gluconeogenesis or ATP synthesis. The significance of this mechanism is especially important for tumor cells that coexist under conditions of both normoxia and hypoxia. Thus, glycolysis-dependent tumor cells under hypoxic conditions export lactic acid to the extracellular space, where it is captured by well-oxygenated tumor cells and used to produce ATP through cellular respiration. Anaerobic glucose oxidation allows cells to adapt well to the hypoxic environment found in tumors, but it is also characteristic of ancient living organisms that lived in anoxic environments prior to the environmental oxygenation event that occurred about 800 million years ago. The Warburg effect observed in tumors can be seen as a return to the ancient hypoxic roots of early multicellular life [13] and maybe to the metabolic profile of the FUCA (the first universal common ancestor). It can be said that malignantly transformed cells form their tissue microenvironment in such a way as to recreate “atavistic” niches favorable for tumor growth in which they can win in competition with healthy cells.

### 2.4. Lactate Promotes Angiogenesis during Tumor Growth

Lactate produced by tumor cells promotes endothelial cell activation and angiogenesis. Interestingly, lactate may act through both HIF (hypoxia-inducible factor)-dependent and HIF-independent mechanisms, and all of them involve the import of lactate by tumor microenvironment cells through the MCT1 transporter and the subsequent inhibition of prolyl hydroxylase. The latter protects HIF from proteasomal degradation and promotes the activation of proangiogenic IL-8 or induces VEGF expression to promote vascular growth. Thus, HIF-1α can be stabilized, not only under hypoxic conditions but also under normoxic conditions, with the help of lactate. Lactate accumulation allows tumor cells to activate the expression of procarcinogenic genes due to the transcriptional activity of HIF-1α, regardless of oxygen supply [14].

### 2.5. Lactate Promotes Cell Migration, Metastasis, and Secretion of Tumor Exosomes

Cell migration is a mandatory step in the processes of carcinogenesis and metastasis. Lactate is one of the factors required for endothelial cell [22] and glioma cell [23] migration by inducing the transcription of transforming growth factor b2 (TGF-b2). Furthermore, the release of lactate and protons from tumor cells into the extracellular environment (microenvironment) and, as a result, the acidification of the microenvironment promote the secretion of exosomes by tumor cells [24]. Exosomes are microvesicles that contain microRNAs, enzymes, structural proteins, and other molecules required for the metabolic reprogramming of cells in the microenvironment. Captured exosomes can induce epigenetic changes or carry oncogenes and onco-microRNAs, which contribute to the rearrangement of the metabolism of the cells surrounding the tumor and further tumor growth.

The mechanisms of lactate’s influence on carcinogenesis induction and tumors progression suggest the activation of the “hyaluronic system” in tumor-associated fibroblasts and the stimulation of VEGF and HIF-1α [22]. It has been shown that HIF-1α is not the only signal that induces the expression of glycolytic enzymes and therefore regulates glycolysis. Metabolites of glycolysis (e.g., pyruvate) and TAC (e.g., oxaloacetate, succinate, and fumarate) can also activate HIF-1α when they accumulate in the cell (because of violation of the ratios between glycolysis and oxidative phosphorylation) [25]. Among other things, pyruvate prevents the aerobic degradation of HIF-1α and enhances the expression of HIF-1α-activated genes, including erythropoietin, VEGF, glucose transporter-3 (GLUT-3), and aldolase A [26]. In turn, VEGF expression is one of the key events in angiogenesis and the subsequent hematogenous migration of cancer cells [27].

### 2.6. Lactate Stimulates Cells Reprogramming to a Stem-Like State

An important feature of a malignant tumor is the existence of a subpopulation of cancer stem-like cells (CSC) characterized by an increased self-renewal capacity and the ability to reconstitute the tumor along with the formation of tumor heterogeneity [28]. Several studies have demonstrated that the reprogramming of differentiated cells to a stem-like state requires the activity of certain transcription factors, e.g., MYC, SLUG, and SOX2 [29]. The Warburg effect in tumors is provided by increased activity of lactate dehydrogenase A (LDHA), which converts pyruvate to lactate. The deregulation of LDHA is observed in many cancers, including prostate, breast, hepatocellular, and gastrointestinal cancers [30,31,32]. At the same time, the inhibition of LDHA suppresses tumor formation and progression, indicating an important role for LDHA in the malignant transformation process [33]. It was shown that the overexpressed LDHA elevates the “stemness” properties of CSCs and enhances spheroid formation in hepatocellular carcinoma [34]. In the work of Cui et al. [35], a molecular pathway that promotes the progression of breast cancer by acting directly on CSCs was investigated. It was demonstrated that elevated lactate production through induced LDHA activity directs USP28-mediated deubiquitination and the stabilization of MYC, thereby promoting stem-like traits in breast cancer. Furthermore, during the study of breast cancer stem cells (BCSCs), it was assumed that LDHA could promote the ubiquitination and endocytosis of E-cadherin to facilitate the transformation of CSCs from epithelial (proliferative)-type BCSCs to mesenchymal (quiescent) BCSCs with high metastatic capacity. The maintenance of the stemness of breast cancer cells is also provided by activating key inflammatory pathways in tumor-associated macrophages (TAM) that constitute up to 50% of tumor-infiltrating cells. It was demonstrated that LDHA could prevent the infiltration of antitumor immune cells (CD4^+^/CD8^+^ T cells) and enhance the accumulation of the protumoral immune cells (bone marrow-derived immunosuppressive cells and TAMs) in the tumor microenvironment (TME) [36,37].

Furthermore, lactate has been shown to promote the CSC phenotype and tumor-initiating capacity of CSCs in colorectal carcinoma cells [38], oral squamous cell carcinoma [39], pancreatic adenocarcinoma cells [40], and some other cancers. Mechanistically, the SCS phenotype of cancer cells may be advocated through the high activity of MCT1, which is the uptake transporter of lactate and is reliably hyperexpressed in many human cancers [10]. Many studies indicated that lactate entering cancer cells through MCT1 can be converted into pyruvate, which then enhances the oxidative phosphorylation, serving as both an energy supply and a signal to transmit upstream of many stem cells pathways, such as the Wnt and p38/MAPK signals [41]. Moreover, malignant cells may instruct the normal stroma cells to increase the production of lactate, thus providing the necessary energy-rich microenvironment for facilitating tumor progression. This phenomenon was called the “Reverse Warburg effect” and was shown to occur in many types of cancers (Figure 1). In this scenario, lactate released from stromal cells enters tumor cells through MCT1 and is used as an energy-rich substrate as well as for gene expression regulation. Subsequently, increased lactate concentrations lead to changes in the NAD^+^/NADH ratio, affecting cells’ redox potential. On the one hand, this can lead to increased reactive oxygen species production (through the stimulation of respiratory electron transport chain activity), and on the other hand, this can lead to sirtuin activation. Sirtuins are proteins with deacetylase activity that are involved in transcriptional regulation and therefore control processes such as apoptosis, inflammation, stress resistance, mitochondrial biogenesis, energy efficiency after caloric restriction [42,43], and aging [44]. Furthermore, as was shown using the ^13^C-glucose tracer, lactate can directly influence the chromatin state through the lactylation of histone proteins and DNA transcription factors [45], for instance, in conditions of increased levels of acetyl-CoA, lactate acetylation (i.e., H3K27Ac), and lactylation (i.e., H3K18 la) at the loci of pluripotency genes (such as Oct4, Sox2, Klf4, and c-Myc), thus facilitating cellular reprogramming to a pluripotent phenotype [46].

### 2.7. Lactate Is a Universal Signaling Molecule

The ability of lactate to act as a signaling molecule has been known since the discovery of the lactate receptor, GPR81, which is expressed in adipocytes and muscle, immune, nerve, and cancer cells. The GPR81 receptor belongs to a type of G-protein-associated receptor, the HCAR subfamily, while the HCAR1 subtype to which GPR81 belongs is considered the most evolutionarily conserved of all subtypes of HCAR receptors [47]. The activation of GPR81 receptors in adipose tissue leads to the inhibition of lipolysis in adipocytes, indicating a synergistic effect of lactate with insulin and its potential association with the development of obesity [48]. Thus, in experiments in mice, it was shown that mice with the gene encoding GPR81 knocked out gained weight to a much lower extent when kept on a high-lipid diet compared to wild-type mice [49].

Lactate has important functions in the nervous system. The extensive investigations of GPR81’s functions in the brain and retina have revealed many new mechanisms of the action of lactate and its role as a signaling molecule in the processes of angiogenesis in the nervous system, the regulation of neuronal excitation, and neuroprotection [50,51].

Several studies were devoted to investigating the role of lactate in inflammation. It has been shown that, in conditions of inflammation, GPR81 expression in adipocytes and endothelial cells is reduced, while the expression of GPR81 in immune cells is associated with protection against inflammation and the suppression of the innate immune response [52]. Interestingly, the activation of GPR81 in the myometrium during pregnancy reduces its LPS-induced inflammation and therefore the risk of preterm birth and infant mortality [53]. The role of GPR81 in the suppression of innate immunity has been confirmed by several studies. For example, in the work of Ranganathan et al. [52], it was found that GPR81 activation in intestinal dendritic cells and macrophages leads to the induction of T-regulatory cells secreting IL-10 and blocking proinflammatory Th1/Th17 cells that suppress inflammation in the intestine.

Based on the many functions of lactate as a signaling molecule, one should expect its important participation in the process of carcinogenesis. Indeed, in recent years there has been more and more evidence of the critical role of lactate-activated signaling pathways through binding to GPR81 in the regulation of tumor growth (Figure 2).

The expression of GPR81 in malignant tumors was first demonstrated in 2014 [54]. The receptor has been shown to be activated in many types of neoplasms, despite its low expression in benign cells from the same tissues. For instance, it was revealed that the downregulation of GPR81 in pancreatic cancer cell lines significantly reduced the expression of the lactate transporters MCT1 and MCT4, while the knockdown of GPR81 resulted in a significant decrease in mitochondrial activity and a marked increase in cell death [54]. The importance of GPR81 for tumor cell survival has also been demonstrated in breast cancer. Among all types of tumors, GPR81 is mainly expressed in breast cancer, especially cells positive for the estrogen receptor (ER), where it increases the production of proangiogenic amphiregulin through a PI3K/Akt/cAMP-dependent pathway [55]. In addition to breast and pancreatic cancer, the pro-oncogenic function of GPR81 is also associated with the development of hepatocellular carcinoma, cervical squamous cell carcinoma, and lung cancer [20]. Mechanistically, GPR81 activation promotes DNA repair (due to the activation of DNA repair protein expression—BRCA1, nibrin, and DNA-dependent protein kinases) and resistance to chemotherapy [56]. At the same time, an increase in DNA repair and drug resistance is associated with the activation of PKC-ERK signaling pathways that obey GPR81.

In addition to regulating angiogenesis, DNA repair, and chemoresistance, GPR81 activation is also required to evade an immune response. For instance, GPR81 activation increases the expression of membrane-bound PD-L1 on the surface of lung cancer cells [20]. Mechanistically, downstream signaling through the Gi/o protein results in the translocation of TAZ/TEAD to the PD-L1 promoter and the subsequent induction of PD-L1 expression in vitro. 

Tumor-derived lactate can be considered a universal promoter of tumor growth due to its ability to induce autocrine effects, including the activation of GPR81 expressed on tumor cells themselves. Interestingly, lactate itself induces the expression of GPR81 in cancer cells by transcriptional activation involving the Snail/EZH2/STAT3 transcription complex [57].

On the other hand, lactate is a paracrine regulator that binds to receptors in cells of the tumor microenvironment. It was found that GPR81 is expressed in tumor-infiltrating immune cells or adipocytes, which make up a significant mass in the mammary gland and are therefore involved in the creation of the tumor microenvironment in breast cancer [58]. Since adipose tissue performs an important endocrine function, the activation of GPR81, including through tumor-derived lactate, can lead to the release of many cytokines and other regulatory factors that affect angiogenesis, vascularization, and tumor growth. However, the exact mechanisms and effects of lactate binding to its receptors remain to be elucidated.

Malignant cells are usually characterized by the presence of a set of distinctive features that arise as a result of the loss and gain of certain functions in cells. It is noteworthy that these signs do not occur ab initio; new functions that tumor cells acquire have always been encoded in the genome, since they play key roles in processes such as ensuring genetic diversity, embryogenesis, wound healing, etc. However, before tumor transformation, they are in a latent state [59]. 

Looking at how tumor cells use an evolutionarily ancient metabolite, lactate, and the process of glycolysis for their proliferation, growth, and metastasis, we can observe the action of ancient metabolic mechanisms for the adaptation and further evolution of cancer cells.

### 2.8. Lactate Affects the Microenvironment of Tumor Cells and Ensures the Formation of Other Oncometabolites

In addition to high lactate content, a hallmark of the tumor metabolic microenvironment is the presence of adenosine, vascular endothelial growth factor (VEGF), phosphatidylserine, and high levels of extracellular K+ as well as acidosis and hypoxia [60]. While studying the involvement of lactate in the synthesis of TAC intermediates in two mouse models of lung cancer, it was shown that its contribution was higher than that of glucose. Using intravenous infusions of ^13^C-labeled nutrients, researchers showed that the circulating flux of lactate is higher compared all other metabolites and exceeds that of glucose in human lung tumors [61]. It was also recently demonstrated that ^13^C-pyruvate is mainly directed to lactate production, which is associated with tumor progression and metastasis [62]. However, in addition to glucose and pyruvate, glutamine can also be a lactate precursor [63]. At the same time, the volume of consumption and metabolism of glutamine is associated with the stabilization of HIF1-α by lactate itself. It is known that HIF1-α transactivates the c-MYC proto-oncogene, which is one of the main regulators of glutaminolysis and is overexpressed in most tumors [64].

Tumor cells can improve their microenvironment in terms of amino acid composition, and this is facilitated by the high content of Na^+^ inside the tumor [65,66]. An increase in the concentration of Na^+^ in the tumor leads to an increase in glucose transport to tumor cells and tumor-associated stromal cells, which contributes to the development of the Warburg–Crebtree effect and the accumulation of lactate. This shift in metabolism towards a more glycolytic phenotype confers numerous advantages for tumor cell survival, including survival in the hypoxic tumor core, and is associated with rapid cell proliferation. For instance, in an experiment on the HeLa cell line, it was shown that the more Na^+^ in the tumor, the higher the production of lactate [67].

It is also known that the activity of Na^+^/K^+^-ATPase can regulate the expression of glycolytic enzymes and mutations in GPR35, a G-protein that is a receptor for kynurenic acid, increases the activity of Na^+^/K^+^-ATPase, and increases the rate of glycolysis. Conversely, the inhibition of Na^+^/K^+^-ATPase reduces the expression of the hypoxia sensor HIF-1α, preventing the activation of glycolysis. It is important to note that any metabolic reprogramming will have an inevitable effect on mitochondrial metabolism. Thus, in cells, including tumor cells, there is a mitochondrial Na^+^/Ca^2+^(Li^+^) exchanger (NCLX), which exchanges mitochondrial Ca^2+^ for cytoplasmic Na^+^ (or Li^+^). Thus, NCLX uses Na^+^ transport to fine-tune mitochondrial Ca^2+^, thereby regulating mitochondrial metabolism, redox homeostasis, and ATP and reactive oxygen species (ROS) production [68]. 

It is quite logical that the consequence of the hypoxic state of tumors is acidosis, an increase in the levels of lactate and adenosine, and an increase in the catabolites of various amino acids and lipids (Figure 3). Many researchers believe that hypoxia is the factor that contributes to the formation and maintenance of CSC stemness, the formation of genetic instability, and tumor heterogeneity. Hypoxia is responsible for malignant progression, sustained neo-angiogenesis, and the development of chemoresistance as well as metabolic reprogramming. At the cellular and molecular level, hypoxia promotes the selection and clonal expansion of cells with inactivated p53 [69].

## 3. Discussion

As highlighted in this review, lactate is a unique oncometabolite that defines mechanisms based on the metabolic and epigenomic reprogramming of transforming cells, thus promoting tumor progression and adaptation to the influence of the microenvironment. It is not surprising that lactate quantification is used for the prediction of tumor grades and prognosis in different cancers [70]. However, despite the abundant evidence of the procarcinogenic effect of lactate and lactatemia being used as a marker of an unfavorable prognosis for the course of oncological disease, in our opinion not everything is so unambiguous in assessing the role of lactate in tumor growth. Currently, researchers are coming to understand the protective function of lactate and the possible compensatory mechanism of lactatemia to overcome stressful conditions in the body. For example, sodium lactate infusions have been found to improve recovery after prolonged exercise in athletes [71]. An intravenous administration of lactate (in the form of sodium lactate) provides an energy substrate, spares glucose, and has a slightly alkalizing effect on blood pH. Lactate is suggested to be used as an anti-inflammatory agent for infections, injuries, burns, hepatitis, pancreatitis, and sepsis and as a wound healing agent, glycemic regulator, source of energy in case of myocardial injury, and neuroprotective agent in cerebral ischemia [9]. Thus, a high level of lactate can compensate for the effects of acute or chronic hypoxia, inflammation, or injury. Lactatemia, observed in many tumor diseases, is possibly associated with the activation of ancient evolutionary defense mechanisms aimed at combating metabolic disorders. However, tumors began to use this mechanism for their own purposes—the acquisition of stem properties, rapid proliferation, and metastasis. It remains to be seen whether this phenomenon can be called a case of antagonistic pleiotropy.

## 4. Conclusions

Summarizing current knowledge about the functions of lactate, the following key positions can be distinguished:The lactate oxidation process, together with glycolysis, maintains the redox potential in the cytosol and mitochondria [72], which in turn is a very important evolutionarily conservative homeostatic constant.Lactate is a universal and ancient signaling molecule. Lactate regulates protein expression, the secretion of signaling molecules, cell proliferation and differentiation, immune surveillance, the inflammatory response, the functioning of transporters and receptors, lipolysis, gluconeogenesis, the content of polyADP-ribose, and the regulation of prolyl hydroxylases, and consequently it is involved in the remodeling of the ECM.Lactate is a universal fuel molecule for rapidly growing tissues and activated cells [73]. Stemness and hypermetabolism are always provided by high lactate production.Lactate has a neuroprotective function during hypoxia [74], which is unconditional and is the most important evolutionary mechanism.Lactate provides stemness and unlimited cell growth, which is used by malignant tumors for their initiation and progression.High lactate production may have protective effect that compensates for pathological conditions such as hypoxia, inflammation, injury, and tissue destruction.

## 5. Summary

○Lactate is not only a universal fuel molecule and the main substrate for gluconeogenesis, but it is also one of the most ancient metabolites with a signaling function, which has a wide range of regulatory activity. ○Lactate regulates key metabolic processes such as proteins expression, the secretion of signaling molecules, cell proliferation and differentiation, immune surveillance, the inflammatory response, etc.○Lactate provides stemness and unlimited cell growth, which is used by malignant tumors for their initiation and progression.○Lactatemia, observed in many tumor diseases, is possibly associated with the activation of ancient evolutionary defense mechanisms aimed at combating metabolic disorders. However, tumors began to use this mechanism for their own purposes—the acquisition of stem properties, rapid proliferation, and metastasis.

## Figures and Tables

**Figure 1 cancers-14-04552-f001:**
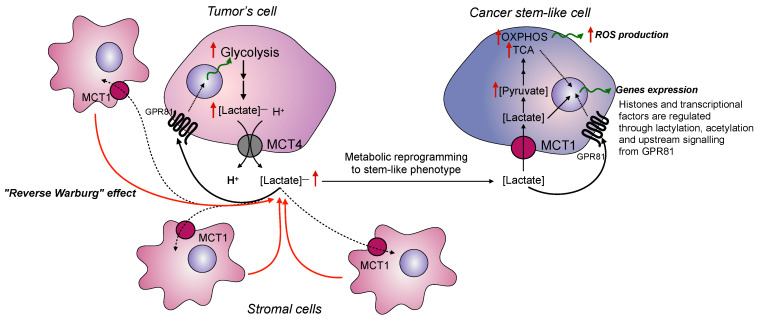
The role of lactate in cancer stem-like phenotype promotion. The accumulation of lactate in the TME leads to its metabolic reprogramming and affects the production of lactate by stromal cells (reverse Warburg effect). Furthermore, lactate may directly influence gene expression through the lactylation of histones and transcriptional factors.

**Figure 2 cancers-14-04552-f002:**
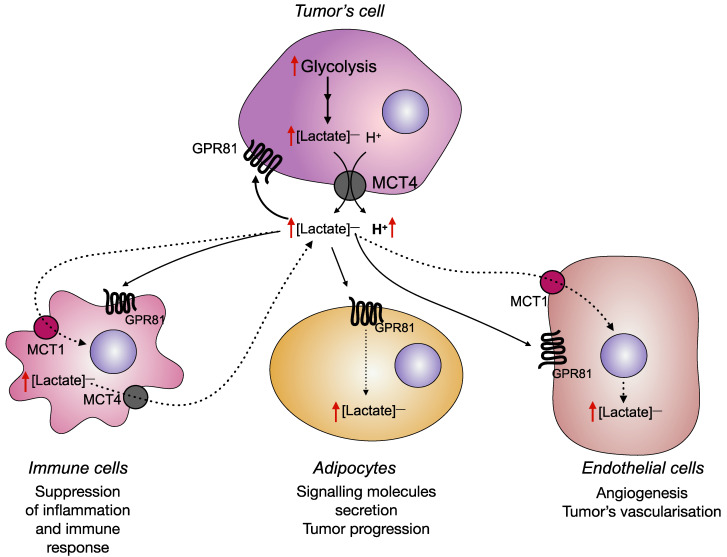
Lactate signaling function in the malignant transformation process. The intensification of glycolysis in tumor cells leads to high lactate production; lactate anions along with protons are cotransported through MCT4 transporters in the extracellular medium that provides acidification of the tumor microenvironment and high extracellular lactate concentration. By binding to GPR81 receptors on different cell types, lactate promotes the expression of pro-oncogenic genes, immune suppression, angiogenesis, tumor cell proliferation, and some other effects.

**Figure 3 cancers-14-04552-f003:**
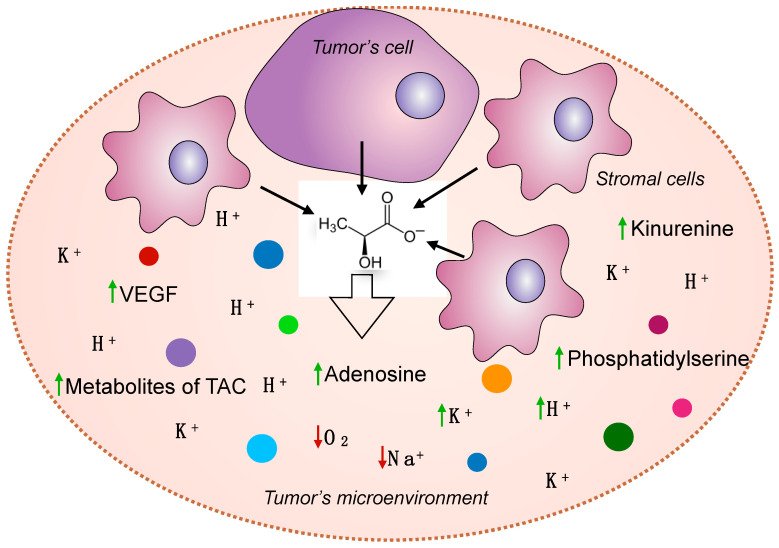
Metabolic reprogramming of the tumor microenvironment due to the Warburg effect. Increased lactate concentrations in the TME promote its acidification and the accumulation of different metabolites, whereas the concentrations of oxygen and sodium ions decrease.

## Data Availability

Data are contained within the article.

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
