# Peer review of "Evolutionary View on Lactate-Dependent Mechanisms of Maintaining Cancer Cell Stemness and Reprimitivization"

_cancers, 2022, doi:10.3390/cancers14194552_

Round 1

Reviewer 1 Report

The review article entitled „Evolutionary view on lactate-dependent mechanisms of maintaining cancer cells stemness and reprimitivization„ provides an insight into an important topic of oncology relating to the evolutionary principle of tumor development.

By contributing to the reprogramming of cancer cells towards those traits resembling "ancestral unicellularity", the metabolism has an importat role. In this context, lactate is not longer regarded as a waste product of the forced glucose metabolism in tumors (Warburg effect). Instead, lactate has been recognized as an important mediator in the switch from quasi multicellularity to unicellularity as seen in cancer. Thus, lactate can act as driver of immune evasion, adapted cancer cell growth and self renewal potential.

In their review article, the authors particularly address the role of the lactate receptor GPR81 in mediating the effects of lactate on cancer cells as well as the impact of lactate on the microenvironment  - i.e. by skewing immune cell functions or angiogenesis.

While these aspects are discussed quite well, „stemness“ as a central topic of the review - at least as one would expect from the title - is not addressed, at all. Thus, the author did not mention the direct effects of lactate on selfrenewal capacities and the stem cell-like phenotype of cancer cells, though many studies have shown the role of lactate in CSCs (i.e. in colorectal, pancreatic, oral cancers). Moreover, the authors did not pay attention to the role of the lactate import in cancer cells - through MCT1 - which essentially contributes to the malignancy of many cancers. Indeed, a plethora of studies have shown that lactate released by glycolytic cancer and stromal cells feeds a subset of cancer cells undergoing oxPhos metabolism (reverse Warburg). In the the latter, lactate serves not only as fuel but directly impacts on the phenotype of CSCs and reprogramming, i.e. by essentially driving epigentic alterations (histone modifications such as acetylation and lactalytion).

In view of the lack of all these important aspects on the role of lactate in stemness, reverse Warburg and epigentic alterations, the authors need to substantially impove their review article. It would be also helpful to provide an improved cartoon (the overview in figure 1 is just too simple) that provides more details on the cellular actions of lactate - including those in reverse Warburg cells (MCT1 driven import) and epigenetic effects.

Author Response

Dear Reviewer!

Thank you for your attentive consideration of our review and your valuable critical comments. 

Point 1: The authors did not mention the direct effects of lactate on self-renewal capacities and the stem cell-like phenotype of cancer cells, though many studies have shown the role of lactate in CSCs (i.e. in colorectal, pancreatic, oral cancers).

Response 1: We agree with the comment of the reviewer and significantly revised our review. The subsection “2.6. Lactate stimulates cells reprogramming to a stem-like state” was added. In this subsection we discuss how activation of key inflammatory pathways in tumor infiltrating immune cells, LDHA activation, MCT1 hyper expression and some other factors direct tumor cells get on the road of reprogramming to obtain stemness capabilities.

Point 2: The authors did not pay attention to the role of the lactate import in cancer cells - through MCT1 - which essentially contributes to the malignancy of many cancers. Indeed, a plethora of studies have shown that lactate released by glycolytic cancer and stromal cells feeds a subset of cancer cells undergoing oxPhos metabolism (reverse Warburg). In the the latter, lactate serves not only as fuel but directly impacts on the phenotype of CSCs and reprogramming, i.e. by essentially driving epigentic alterations (histone modifications such as acetylation and lactalytion).

Response 2: The authors agree with the comment of the reviewer and significantly revised the review. The subsection “2.6. Lactate stimulates cells reprogramming to a stem-like state” was added. In this subsection we have paid attention to the role of the lactate import in cancer cells through MCT1, as well as discuss a Reverse Warburg effect and the role of lactate in epigenetic modifications leading to changes in cells metabolism.

Point 3: It would be also helpful to provide an improved cartoon (the overview in figure 1 is just too simple) that provides more details on the cellular actions of lactate - including those in reverse Warburg cells (MCT1 driven import) and epigenetic effects.

Response 3: The authors agree with the comment of the reviewer. We added one more figure describing the reverse Warburg effect and lactate influence on acquiring stem-like phenotype by cancerous cells, and also we have modified figure 1 (it has become figure 2) specifying the effect of lactate in different cells.

Reviewer 2 Report

Shegay and colleagues point out comprehensively the role of lactate from many points of view. Although the topic is highly consumed and not new at all. But I found out that it’s well written in nice way moreover they covered all aspects in one review which is unique and needed nowadays. I highly recommend publication but although I have some minor comments:

1) Its would add much more value to this review if you add the importance of lactate to differentiate between benign tumor and malignant tumor

2) Highlight more the role of lactate in redox pathway

3) Also would be add on to go deeper in the metabolic changes (other metabolites) due to lactate changes.

4) If you could add the Techniques used to measure lactate from different type of samples

Thank you and good luck!

Author Response

Dear Reviewer!

Thank you for your attentive consideration of our review and your valuable critical comments. Please see the attachment and edited manuscript where we tried to eliminate the shortcomings of the work.

Reviewer 3 Report

Dear authors,

thanks for the review. I knew that I was right when I agreed reviewing it. It is the right appetizer to dig deeper.

Well written - with some small spelling errors - and easy to understand

Author Response

Dear Editor! Thank you for your review and for the appreciation the importance of the topic. We will attentively check the manuscript to eliminate spelling errors.

Thank you very much!

Reviewer 4 Report

The text reviews the role of lactate in the development of tumor cells, which presents an update on the biological meaning that this metabolite represents in tumor development. The text is easy to read and efficiently manages to explain the central argument.

Author Response

Dear Editor! 

Thank you for your review. We really appreciate your high evaluation of our work. The text of the manuscript wil be checked to eliminate spelling errors and according to the critical comments of other reviewers.

Thank you very much!

Round 2

Reviewer 1 Report

The authors significantly improved their review article. In particular, the effects of lactate on stemness and epigenetic alterations as well as the context of its MCT1 driven import (reverse Warburg) are discussed now sufficiently. By adding this important aspect to their discussion of the environmental and GPR81 mediated lactate effects, the authors now provide a more complete update on the different roles of lactate in tumor development and growth. Moreover, the paper is now accompanied by 3 figures that illustrate the different modes of action of lactate more comprehensively. 

Therefore, I recommend to publish this review article in Cancers.